# Preparation and Photodegradation of TiO_2_ Thin Films on the Inner Wall of Quartz Tubes

**DOI:** 10.3390/ijms251910253

**Published:** 2024-09-24

**Authors:** Wei Chen, Qi Yan, Junjiao Yang

**Affiliations:** 1College of Chemistry, Beijing University of Chemical Technology, Beijing 100029, China; 2021210562@buct.edu.cn; 2Analysis and Test Center, Beijing University of Chemical Technology, Beijing 100029, China; yanqi0604@buct.edu.cn

**Keywords:** titanium dioxide, photocatalytic degradation, quartz tube

## Abstract

Titanium dioxide thin films on the inner wall of quartz tubes were prepared in situ by the sol–gel method. Meanwhile, copper and cerium were loaded onto the surface of the titanium dioxide thin films to enhance photocatalytic activity and broaden the range of light absorption. X-ray diffractometer, X-ray photoelectron spectroscopy, scanning electron microscopy, energy dispersive X-ray spectrum, N_2_ gas adsorption, UV diffuse reflectance spectroscopy, electron paramagnetic resonance, photoluminescene spectroscopy, and so on were used to characterize the structure, morphology, chemical composition, and optical properties of the prepared photocatalyst. Methylene blue (MB) was used as a simulated organic pollutant to study the photocatalytic performance of the photocatalyst, which was a translucent, structurally stable, and reusable high-efficiency photocatalytic catalyst. Under UV lamp irradiation, the MB photodegradation efficiency was 94.5%, which reached 91.2% after multiple cycles.

## 1. Introduction

With globally increasing industrialization, the amount of pollution is increasing annually; similarly, the energy needed to treat this waste is also increasing. Therefore, it is crucial to find energy-efficient solutions for treating solid and liquid wastes, as well as exhaust gases [1,2,3]. Upon irradiation with a certain wavelength of light, semiconductor materials can convert that energy into chemical energy. Thus, they can promote the oxidation of organic pollutants and degrade them [4,5,6]. Semiconductor catalytic degradation has the following advantages: (1) fast (minutes or hours) and complete degradation; (2) diversity of species degraded, including aromatic and chlorinated organic compounds, which are generally difficult to degrade; (3) low energy consumption and mild reaction conditions that proceed when exposed to ultraviolet rays or sunlight; and (4) thorough mineralization and no secondary pollution. In addition to these advantages, titanium dioxide (TiO_2_)-based materials are non-toxic, chemically stable, and highly photocatalytic [7,8,9,10,11,12]. Currently, there are two main types of TiO_2_ photocatalysts: powders [13] and thin films [14]. Powdered TiO_2_ has a large specific surface area and high photodegradation efficiency. However, it is difficult to recover, has a low reusability rate, and has low economic benefits. The thin-film method involves coating a carrier substrate with TiO_2_, but it can easily peel off, and the reuse rate is also low. Many researchers have coated TiO_2_ on the surfaces of metal [15], glass [16], and ceramic [17] materials. However, the catalysts prepared by this method transmit light poorly, thus lowering the photocatalytic degradation efficiency. As an alternative, this study proposes coating the inner wall of quartz tubes with TiO_2_, yielding a high photodegradation efficiency and reusability. Thus, we can conserve energy and protect the environment.

TiO_2_ is a typical n-type semiconductor widely used because of its nontoxic nature, high photocatalytic activity, and excellent chemical stability [18,19,20]. There are three main types of TiO_2_ crystals: rutile, anatase, and brookite [21]. In general, anatase and rutile TiO_2_ can transform into each other at different calcination temperatures. Wang Bin et al. found that in the early stages of TiO_2_ crystal cell growth, the TiO_2_ grains are small and have significant lattice defects. However, increasing the calcination temperature improves the crystallinity of TiO_2_. The carrier substrate used also affects the growth of TiO_2_ grains [22]. TiO_2_ is categorized as anatase or rutile depending on grain size. When the grain size is less than 10–16 nm, the grains usually exist in anatase form [23]. Brookite TiO_2_ grains have a higher electron migration rate, lower dielectric constant, and lower density, making them preferable for photodegradation applications [24].

During the photodegradation of TiO_2_, electrons in the valence band absorb light energy and transition to the conduction band, resulting in the formation of holes in the valence band. Photogenerated electrons in the conduction band mainly undergo reduction reactions, whereas the holes in the valence band undergo oxidation reactions [25]. The active substances generated by photocatalysis are mainly hydroxyl radicals and singlet oxygen atoms. The Figure 1 shows a schematic representation of the photocatalytic degradation mechanism. Light can excite electrons in TiO_2_ semiconductors, driving them from the valence band to the conduction band to generate photogenerated electrons. Corresponding photo generated holes are generated in the valence band, which diffuse to the semiconductor surface and react with different reactants on the surface.

To enhance photocatalytic performance, TiO_2_ can be modified and loaded with metal compounds [21]. Doping with metals provides carriers for the transfer of photogenerated electrons, reducing the recombination of photogenerated charge carriers on TiO_2_. Generally, transition or rare-earth metals are used for doping [26]. For example, cerium oxide’s crystal structure contains active sites that make it useful and efficient in waste gas and wastewater purification [27]. Copper oxide helps generate hydroxyl radicals during the photocatalytic process and can also generate hydroxyl radicals under light conditions [28,29]. The co-doping with Cu and Ce of TiO_2_ reduces the bandgap width, expands the absorption range of visible light, and improves the photocatalytic efficiency of photodegradation.

Quartz tubes have good transmission of light, especially short-wavelength ultraviolet light, making them an excellent carrier for TiO_2_ photocatalytic materials. Therefore, quartz tubes were used as carrier substrates to grow TiO_2_ on their inner walls, and load metals were used to reduce the bandgap and improve the degradation efficiency. To achieve this, the effects of varying the reactant concentrations on the growth of TiO_2_ were studied, as well as the influence of the calcination temperature on the crystal structure of TiO_2_ and the influence of the concentration of impregnated metal salts on the loading amount. The high catalytic activity, wide spectral absorption range, and repeatedly use photo-degradation catalysts were preparated.

## 2. Results and Discussion

### 2.1. Catalyst Characterization

#### 2.1.1. Cross-Sectional Morphology

Figure 2 shows a scanning electron microscopy (SEM) image of the cross section of the quartz tube. Figure 2A shows the overall SEM image of the quartz tube. Figure 2B,C show a locally enlarged view of the quartz tube. We observe that TiO_2_ formed a uniform thin film on the inner wall of the quartz tube.

The thickness of TiO_2_ within the quartz tube (seen in Figure 3) increases with a larger volume fraction of the tetrabutyl titanate (TBT) in the reaction solution. The thicknesses of the TiO_2_ film were 5.3, 4.1, 3.2, 2.8, 1.5, and 0.6 μm, and the concentrations of TBT were 15, 12.5, 10, 7.5, 5, and 2.5%, respectively. However, when the TBT concentration reached 10%, the growth rate of the TiO_2_ films decreased significantly, and the effect of increasing the TBT concentration became less significant. In practice, the TiO_2_ film does not need to be excessively thick. If it is, it reduces the transparency of the quartz tube, thereby weakening its photocatalytic ability.

Different thicknesses of TiO_2_ films were prepared by adjusting the amount of TBT. Figure 4 shows the degradation efficiency of 15 ppm methylene blue (MB) by the TiO_2_ thin films of different thicknesses. The degradation of MB was determined after regular time intervals utilizing the following equation: (%dye degradation = (Co − Cf))/Co × 100 = (Ao − Af)/Ao × 100) and was used to determine the kinetics of photodegradation reactions [30,31]. By fitting the experimental MB photo-degradation data to first-order kinetics where “lnCo/Ct” versus “t” was plotted, good R^2^ values for degradation kinetics of MB were obtained, as shown in Figure 4 (left). The degradation efficiency of the quartz tube without a growing TiO_2_ film was almost zero after 45 min illumination, indicating that the quartz lost the ability to photodegrade MB. After loading the TiO_2_ film onto the quartz tubes, the MB degradation ability immediately increased; however, the photodegradation efficiency did not increase with the thickness of the TiO_2_ film. As the thickness of the film increased from 0.6 μm to 2.8 μm, the photocatalytic performance of the film continuously increased. However, when the thickness exceeded 2.8 μm, the photodegradation efficiency plateaued due to a decrease in the transparency of the film, in the utilization rate of light energy, and in the photodegradation efficiency. When the thickness of the TiO_2_ film was 2.8 μm, the corresponding volume fraction of TBT added was 7.5%.

#### 2.1.2. Effects of Ethyl Orthosilicate

TBT hydrolyzes quickly and grows in a lattice-oriented manner. The rate of hydrolysis of ethyl orthosilicate mostly depends on removing the first ethoxy group. The removal of the first ethoxy group was often slower. However, after the removal of the first ethoxy group, the other ethoxy groups were quickly detached and coupled to form silica, which cannot grow in the lattice direction over time. This means that the hydrolysis of ethyl orthosilicate can only generate amorphous silica. Ethyl orthosilicate participates in the hydrolysis of TBT, controls the hydrolysis rate of TBT, and disrupts the directional growth of TBT. This weakens the cohesive strength of the TiO_2_ film and prevents cracking.

#### 2.1.3. Effects of PEG Molecular Weights

Polyethylene glycol (PEG) with different molecular weights has different volumes. When PEG is added to the TBT reaction solution, its hydrolysis generates TiO_2_ that occupies a certain volume. After high-temperature sintering, holes form in the TiO_2_ film, making it a porous structure known as a pore-forming agent, thereby increasing the specific surface area of the film. By adding PEG of different molecular weights (i.e., 400, 800, 1000, 1500, and 2000), the pore size of the film material could be adjusted, which, in turn, changed the specific surface area of the TiO_2_ film. The different specific surface areas of the films had different effects on the photodegradation efficiency. Figure 5 shows the MB photodegradation efficiencies of the TiO_2_ films synthesized from PEG of different molecular weights. The degradation rate increased with an increase in the molecular weight from 400 to 800 to 1000, with the latter yielding the best photocatalytic effect. When the molecular weight increased to 1500 and 2000, the degradation effect began to decrease, possibly because of the excessive pore size, which resulted in a decrease in the specific surface area and catalytic efficiency.

The specific surface area of the TiO_2_ film was determined using the Brunauer–Emmett–Teller (BET) method (see Table 1). As the molecular weight of PEG increased from 400 to 800 to 1000, the specific surface area gradually increased. The molecular weight of PEG gradually increased from 1500 to 2000, but the specific surface area gradually decreased. As the molecular weight of PEG increased, the pore size also increased; however, when the pore size of the TiO_2_ film was too large, the specific surface area decreased. Therefore, PEG with a molecular weight of 1000 was selected as the optimal molecular weight for the experiment.

#### 2.1.4. Effects of Metal Doping on Surface Area

Metal doping affected the specific surface area of TiO_2_ films. Table 2 shows the specific surface area of the films after metal doping. The specific surface area of the film without metal doping was 329.7 m^2^·g^−1^; copper doping can cause a slight decrease in the specific surface area. After Cu doping, larger particles were formed, thereby reducing the specific surface area of the film. This resulted from the formation of larger metal oxide particles after doping, which blocked the pores of some thin films and lowered the specific surface area. Cerium doping increased the specific surface area of the film due to the formation of small particles on the catalyst. When Cu and Ce were doped simultaneously, the specific surface area of the thin films slightly increased.

### 2.2. Thermogravimetric Analysis

Figure 6 shows the thermogravimetric analysis (TGA), which shows that the thermal decomposition temperature was 385 °C. When the temperature exceeded the decomposition temperature, weight loss no longer occurred, indicating that all the organic templates (CTAB, PEG, etc.) in the sample had completely decomposed.

### 2.3. Effects of Calcination Temperature

The calcination temperature of the TiO_2_ films significantly affected the crystal structure of TiO_2_. Figure 7(left) shows the X-ray Diffraction (XRD) patterns of TiO_2_ calcined at different temperatures. The calcination temperature affects the crystal structure of TiO_2_ on the inner wall of the quartz tube, thereby affecting its degradation efficiency. The structure of the brookite is unstable, and the anatase has a higher photocatalytic activity than the rutile. The calcination temperatures ranged from 400 to 500 °C. The diffraction intensity of anatase increased with increasing temperature. At 400 °C, the crystal structure was unstable, and there were many impurity peaks. When the temperature rose to 500 °C, the diffraction peak of the anatase was clear, which was completely stable with the anatase standard card (JCPDS card number of PDF#21-1272). When the temperature rose to 600 °C, the XRD diffraction pattern showed a rutile diffraction peak (see JCPDS card number of PDF#21-1276), indicating that some TiO_2_ had been converted to the rutile. Figure 7(right) also compare the degradation efficiencies of the catalysts toward MB at different calcination temperatures. At 400 °C, the crystallization of TiO_2_ was incomplete, the crystal form was almost amorphous, and the photocatalytic efficiency was poor. At 600 °C, the crystal form of TiO_2_ was basically the rutile, and the catalytic activity of the rutile was worse than that of the anatase, resulting in the lower catalytic efficiency of the TiO_2_ film. The catalyst sintered at 500 °C exhibited the highest activity; therefore, this was the temperature chosen for the optimized experimental conditions.

Figure 8 shows the XRD patterns of TiO_2_ doped with Cu and Ce after calcination at 500 °C. After calcination, pure TiO_2_ exhibits diffraction peaks at (101), (004), (200), (211), and (105), which correspond exactly to the standard anatase. A comparative analysis of the Cu-doped TiO_2_ and pure TiO_2_ samples revealed that the characteristic peaks were almost identical before and after doping. The anatase diffraction peak areas of Ce-doped TiO_2_ at (101) and (200) were more intense than those of pure TiO_2_, indicating that Ce doping increases the stability of anatase TiO_2_. When Ce and Cu were doped simultaneously, a new diffraction peak appeared at 28.61°, which is a result of the combined effect of Ce and Cu on TiO_2_.

### 2.4. SEM-EDS Analysis

Different concentrations of copper nitrate and cerium nitrate solutions were prepared to dope TiO_2_. We studied the relationship between the concentration of the impregnated element and the loading amount and the effect of different doping amounts on the photocatalytic degradation efficiency of MB. Figure 9 shows the scanning electron microscopy energy dispersive spectrum (SEM-EDS) analysis of the Cu and Ce impregnation concentrations versus the loading amounts. As the elemental impregnation concentration increased, the loading amount increased almost linearly. When the Cu impregnation concentration reached 0.2 M, further increasing the impregnation concentration did not result in a significant increase in the loading capacity. Figure 10 shows the degradation efficiencies of MB at different metal impregnation concentrations. As the concentrations of the added elements increased, the degradation efficiency of the catalyst increased toward MB. In this case, too, when the Cu impregnation concentration reached 0.2 M, further increasing the concentration did not significantly increase the degradation efficiency. When the Ce impregnation concentration reached 0.1 M, increasing the concentration reduced the degradation efficiency. As such, the impregnation concentration of Cu was determined to be 0.2 M and of Ce 0.1 M. Figure 11 compares the degradation efficiency of MB by different types of photocatalysts. The rate constant of MB degradation was calculated based on the degradation efficiency data by referring to other works [30].

### 2.5. XPS Analysis

After loading Cu (transition metal) and Ce (rare earth metal) onto the TiO_2_/quartz tubes, the efficiency of MB degradation was significantly improved. Therefore, it was necessary to study and analyze the valence states of these elements. Figure 12 shows the X-ray photoelectron spectroscopy (XPS) of the TiO_2_/quartz tube surface loaded with Cu and Ce. The XPS spectrum of Cu shows two strong absorption peaks at 932.6 eV and 952.2 eV, indicating the presence of Cu 2p^3/2^ and Cu 2p^1/2^. The Cu peaks at 932.6 eV are 932.6 eV, 932.4 eV, and 933.6 eV, indicating that Cu mainly exists in the forms of Cu^0^, Cu^+^, and Cu^2+^, similar to the findings in reference [32,33,34]. High-temperature-sintered Cu with positive divalence transitions to positive monovalent and zero-valent states; Cu_2_O itself can generate hydroxyl radicals, and Cu^1+^ exhibits high chemical activity. Together with TiO_2_, they promote the formation of hydroxyl radicals and improving the degradation efficiency of TiO_2_.

In the 3D orbital spectrum of Ce, Ce 3d^5/2^ is located at 884 eV, corresponding to Ce^4+^, while the peak of Ce3d^3/2^ is located at 902 eV, corresponding to Ce^3+^ [35,36]. The peak areas suggest that the content of Ce^4+^ and Ce^3+^ on the surface were basically identical. The presence of both is suggestive of their mutual conversion, which generates more hydroxyl radicals on the catalyst surface and, thus, improves catalyst performance.

### 2.6. UV Diffuse Reflectance Spectroscopy Analysis

Figure 13 shows the UV-Vis diffuse reflectance spectra (UV-Vis DRS) of TiO_2_ doped with Cu and Ce. In the UV-Vis DRS, due to the need to change the light source (from W lamp to D lamp) in the 450~400 nm wavelength range and switch the grating around 720 nm, the baseline fluctuates between 450~400 nm and 720 nm. From their Tauc plot, the forbidden bandwidth graph can be drawn based on the relationship between the forbidden bandwidth and maximum absorbable wavelength of the Fermi distribution function. The bandgap width of TiO_2_ without metal doping was 3.2 eV, and the maximum absorption wavelength was approximately 390 nm. Compared with TiO_2_ doped by Cu and Ce, the difference in the reduction of the bandgap intensity between the two was not significant. The ability of Cu and Ce to enhance photocatalytic degradation combines with their advantages to enhance the photocatalytic degradation efficiency. For Cu loading, the bandgap width of cuprous oxide is 2.2 eV. Cuprous oxide is a p-type semiconductor, and TiO_2_ is an N-type semiconductor. Together, they can form a P-N heterojunction to promote electron transfer [37] and enhance the maximum absorption wavelength. The advantage of Ce doping lies in its high conductivity and ability to store and release photogenerated holes [38]. Comparing the UV–visible diffuse reflectance spectra of Cu- and Ce-doped TiO_2_, the maximum absorption wavelength significantly increases while the bandgap width decreases to 2.5 eV, which enhances the photocatalytic degradation ability of TiO_2_.

### 2.7. Photoluminescence Spectroscopy Analysis

Fluorescence spectroscopy reflects the recombination rate of photogenerated electron–hole pairs in catalysts [39]. Generally, the lower the line on the photoluminescence spectrum, the weaker the fluorescence intensity, and the lower the recombination rate of the photogenerated electron hole–pairs, which is beneficial for the progress of photocatalytic reactions. Figure 14 shows the photoluminescence spectra of TiO_2_, TiO_2_ doped with Cu, TiO_2_ doped with Ce, and TiO_2_ doped with Cu and Ce. When Cu and Ce are loaded on the catalyst simultaneously, the fluorescence intensity is the weakest, indicating that the recombination rate of the photogenerated electron–hole pairs doped with Cu and Ce is the lowest, and the efficiency of the photocatalytic degradation of the pollutants is higher.

### 2.8. EPR Analysis

Electron paramagnetic resonance (EPR) analysis was conducted on metal-doped TiO_2_/quartz tubes. Dimethyl sulfoxide manganese oxide (DMPO), a radical scavenger, was used as the capture agent to analyze the signal intensity of the hydroxyl radicals in the quartz tube solution under light conditions. The higher the signal intensity, the higher the hydroxyl radical content, and the better the catalytic degradation activity. From Figure 15, we observe that the signal intensity of the Cu-doped free radicals is higher than that of Ce, which also confirms that cuprous oxide can produce hydroxyl radicals under light conditions. Combining the advantages of both metal oxides significantly enhanced the signal intensity of the hydroxyl radicals, playing a positive role in the catalytic degradation ability of the TiO_2_/quartz tubes.

### 2.9. Catalyst Repeatability Stability

A stable catalyst structure and sustained and efficient catalytic activity are important indicators of excellent catalytic performance. Figure 16 shows the results from a repeatability experiment evaluating MB degradation by the TiO_2_ film. The figure shows that the degradation rate of the catalyst for the first use was 94.5%, and the degradation rate for the 10th use was 91.2%, indicating that the prepared TiO_2_ quartz tube has a stable structure and exhibits good catalytic activity after multiple cycles of use.

### 2.10. Degradation Products of MB

To investigate the reaction mechanism of the photocatalytic degradation, the degradation products of MB were detected by liquid chromatography–mass spectrometry (LC-MS). Figure 17 shows the mass spectrum of the MB degradation products, which are explained relative to the fragments in Table 3. MB degradation begins with the removal of the methyl groups. This process may involve the participation of water. In the presence of the catalyst and light, MB reacts with water. After the N-substituted methyl group is removed, it combines with the hydroxyl group of water to form methanol, and the other H in water recombines with the N from MB.

## 3. Materials and Methods

### 3.1. Reagents and Instruments

The compounds used for the catalyst synthesis and MB degradation were all analytically pure reagents (as listed in Table 4). The instruments and equipment used are listed in Table 5.

### 3.2. TiO_2_/Quartz Tube Preparation

#### 3.2.1. Activation of Quartz Tubes

We injected a 500 mm × 1.5 mm (ID) quartz tube with a 1 M sodium hydroxide solution using a syringe, soaked it at room temperature for 2 h, and then rinsed the quartz tube with deionized water until the pH was neutral. We rinsed the remaining water with anhydrous ethanol and finally dried the quartz tube in an oven at 120 °C for 2 h.

#### 3.2.2. Coating of TiO_2_/Quartz Tubes

We injected a 0.5 mM solution of CTAB in water/ethanol (*v*/*v*, 90/10) into an activated quartz tube using a syringe. After soaking for 1 h, the CTAB solution was poured out, and a uniform water film formed on the inner wall surface of the quartz tube.

We transferred 0.75 mL of TBT, 0.5 mL of ethyl orthosilicate, and 0.75 mL of PEG into a test tube using a pipette. Then, 10 mL of the reaction solution was prepared using n-butanol as the solvent. The reaction solution was poured into a constant-pressure funnel, which was connected to the quartz tube with a rubber tube. The quartz tube was placed vertically, the constant-pressure funnel was opened, the flow rate was controlled to one drop per second, and after the reaction solution flowed out, a suction ball was used to blow out the excess reaction solution. We put the quartz tube into an oven and dried it at 120 °C for 1 h. Afterwards, the quartz tube was placed in a muffle furnace, heated to 500 °C at a rate of 5 °C/min, and kept there for 2 h to obtain the TiO_2_/quartz tube.

#### 3.2.3. Modification of TiO_2_/Quartz Tubes

The volume ratio of 0.1 M to the calcined TiO_2_/quartz tubes was 1:1. We soaked them in a 1% aqueous solution for 10 min, blowing out excess liquid. The tubes were dried at 120 °C and then calcined at 500 °C under nitrogen for 2 h in a tubular furnace. We injected 15 ppm of MB into the prepared TiO_2_/quartz tubes and evaluated the photodegradation ability under UV lamp irradiation.

After sintering, the TiO_2_/quartz tubes were soaked in a 1:1 (*v*/*v*) aqueous solution of 0.1 M Ce(NO_3_)_3_ and 0.2 M Cu(NO_3_)_2_ for 10 min. After removing the excess liquid, the tubes were dried at 120 °C and then calcined under nitrogen at 500 °C for 2 h in a tube furnace. We injected 15 ppm of MB into the prepared TiO_2_/quartz tube and verified the photodegradation ability by irradiating with a UV lamp.

## 4. Conclusions

Quartz tubes have excellent transparency and chemical stability. TiO_2_ is fixed to the inner wall of a quartz tube to form a stable, fixed-bed catalyst, which can purify and treat wastewater containing organic pollutants online. TiO_2_ quartz tubes with different film thicknesses were prepared by adjusting the reaction conditions. The thickness of the TiO_2_ film affected its photocatalytic activity. However, a thicker film did not mean a higher photocatalytic activity. When the film thickness exceeded 2 microns, the photocatalytic activity decreased due to a decrease in transparency.

Metal doping can affect the specific surface area of TiO_2_ films, and thereby affect their catalytic activity. The structural stability of a catalyst determines its repeatability, and the photocatalytic degradation rate of the TiO_2_ thin-film catalyst prepared in this study reached 91.4% even after ten cycles of use.

## Figures and Tables

**Figure 1 ijms-25-10253-f001:**
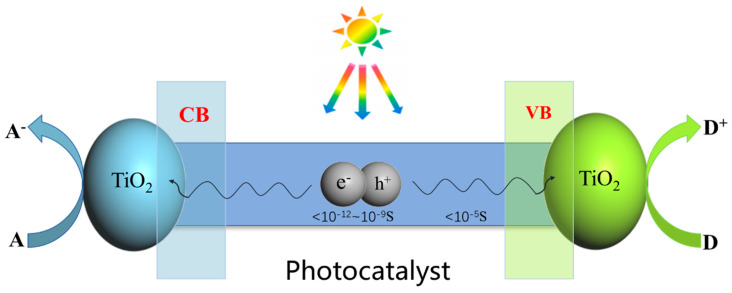
The schematic diagram of photocatalytic degradation of TiO_2_.

**Figure 2 ijms-25-10253-f002:**
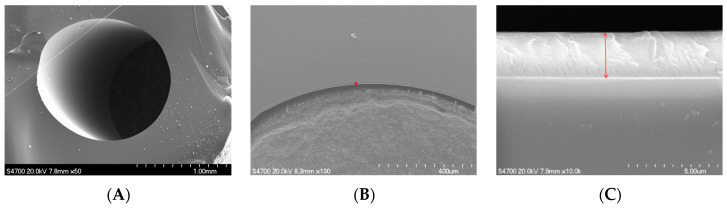
SEM morphology image of the cross section of quartz tube ((**A**–**C**) represent different magnifications). The red mark indicates the thickness of the TiO_2_ film, as shown in (**C**).

**Figure 3 ijms-25-10253-f003:**
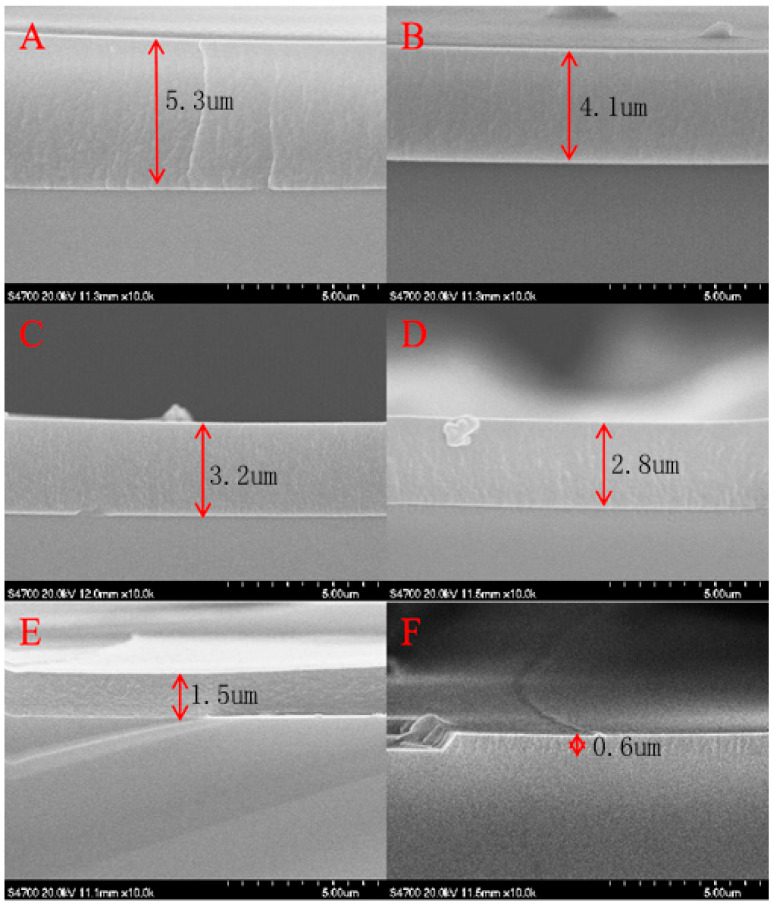
SEM images of the cross-section of quartz tubes with different thicknesses of the TiO_2_ films ((**A**), (**B**), (**C**), (**D**), (**E**), and (**F**) were the concentrations of TBT with 15, 12.5, 10, 7.5, 5, and 2.5%, respectively).

**Figure 4 ijms-25-10253-f004:**
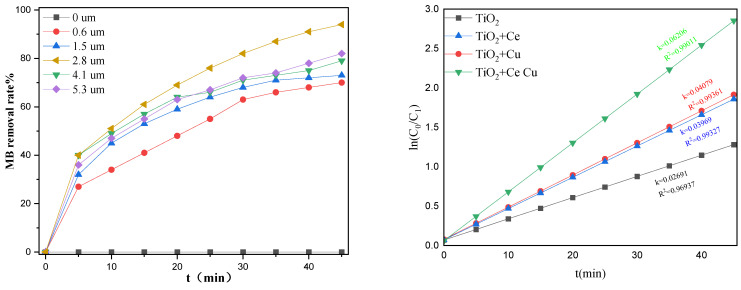
Photodegradation efficiency of MB solution in quartz tubes with different TiO_2_ thicknesses (**right**) and reaction kinetics of dye degradation (**left**).

**Figure 5 ijms-25-10253-f005:**
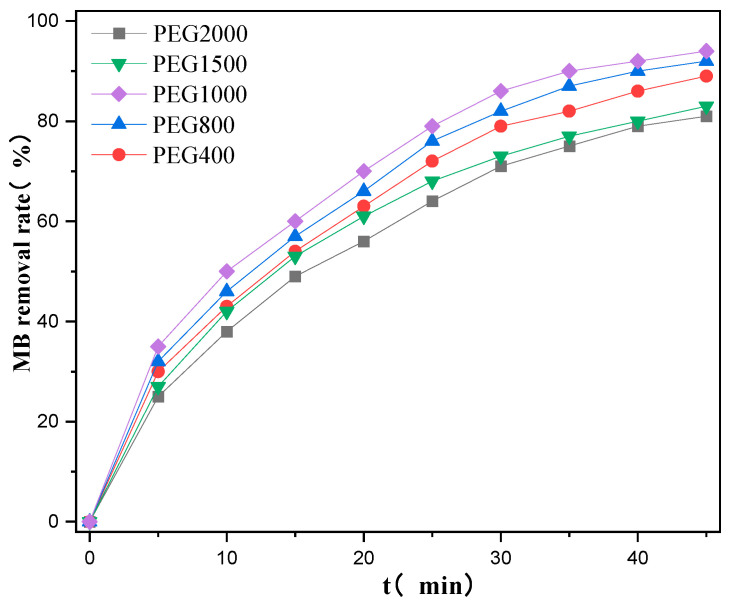
MB degradation efficiency of catalysts with different PEG molecular weights.

**Figure 6 ijms-25-10253-f006:**
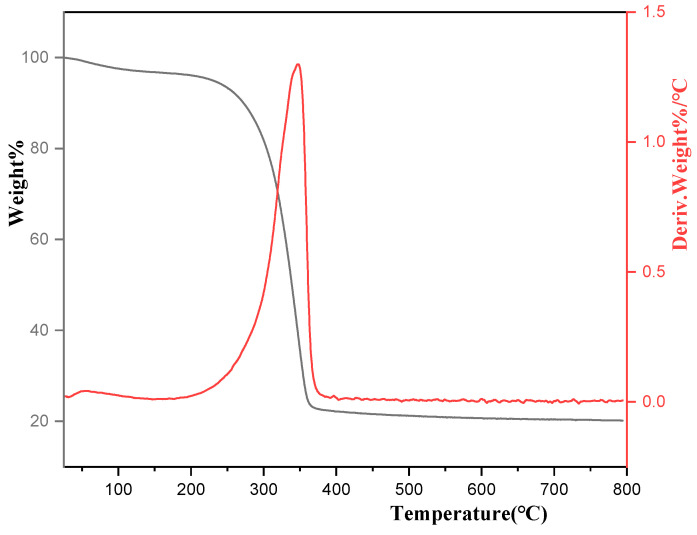
TGA test diagram of sample weight changing with increasing temperature.

**Figure 7 ijms-25-10253-f007:**
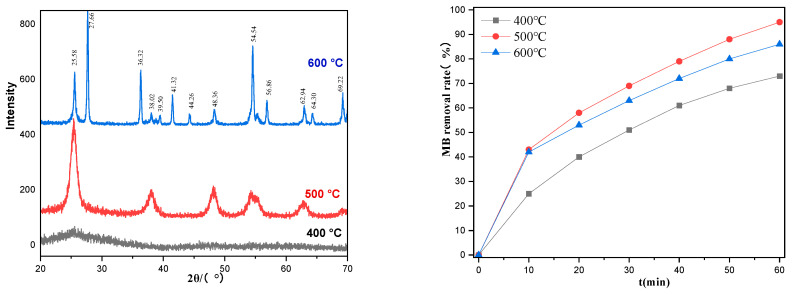
XRD (**left**) and photodegradation plots of catalysts (**right**) at different calcination temperatures.

**Figure 8 ijms-25-10253-f008:**
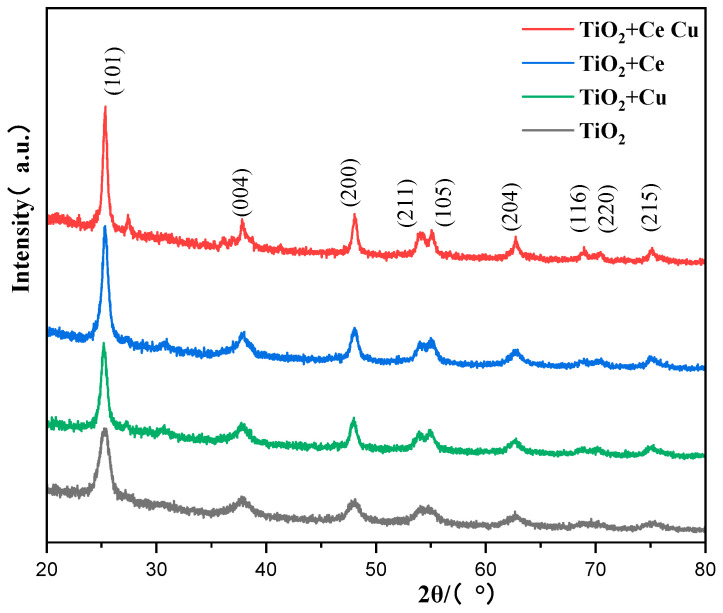
XRD images of the doping metals Cu and Ce under calcination at 500 °C.

**Figure 9 ijms-25-10253-f009:**
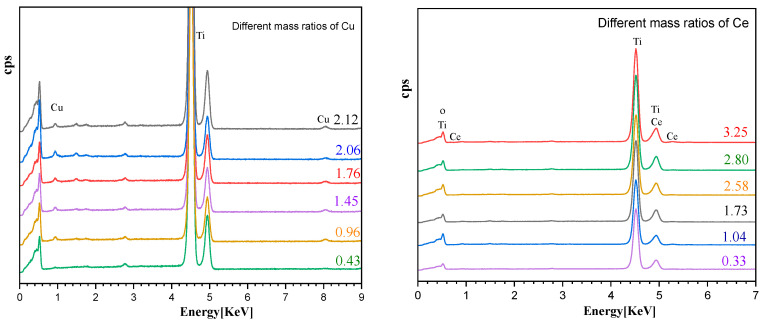
The relationship between element impregnation concentration and loading amount as determined by SEM-EDS. The (**top**) figures show SEM-EDS spectra of elements with different impregnation concentrations. The figures (**below**) show the ratio of the amount of the doping Cu and Ce to Ti at different impregnation concentrations.

**Figure 10 ijms-25-10253-f010:**
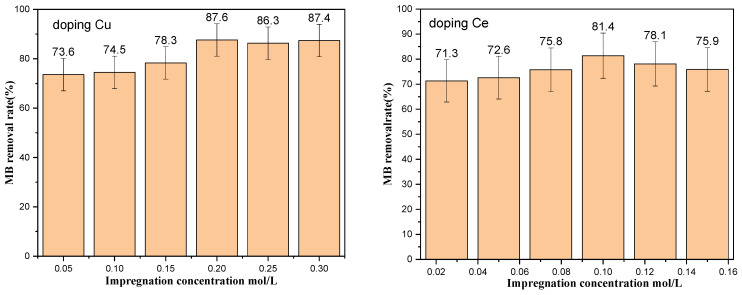
Degradation efficiency of MB at different impregnation concentrations (doping Cu on the (**left**) and doping Ce on the (**right**)).

**Figure 11 ijms-25-10253-f011:**
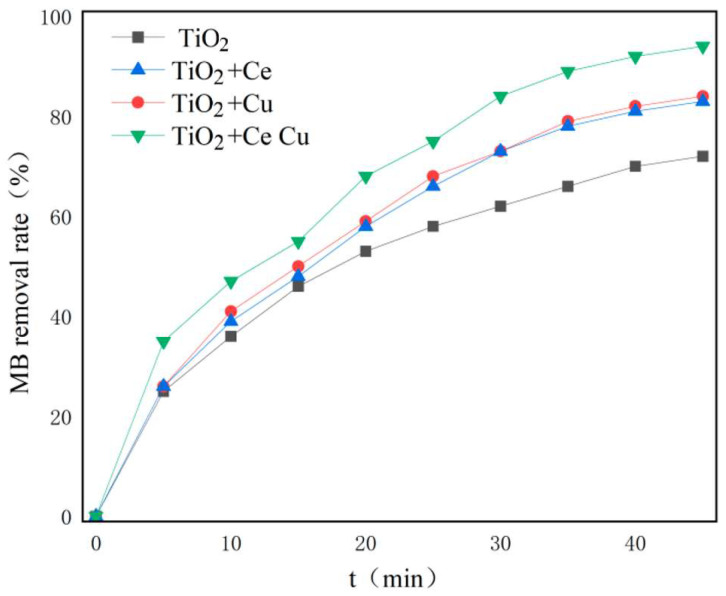
Degradation efficiency of MB for different types of metal-doped TiO_2_.

**Figure 12 ijms-25-10253-f012:**
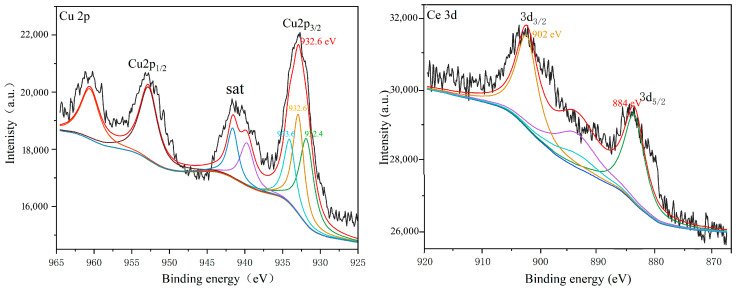
XPS spectra of TiO_2_/quartz tube after Cu and Ce loading.

**Figure 13 ijms-25-10253-f013:**
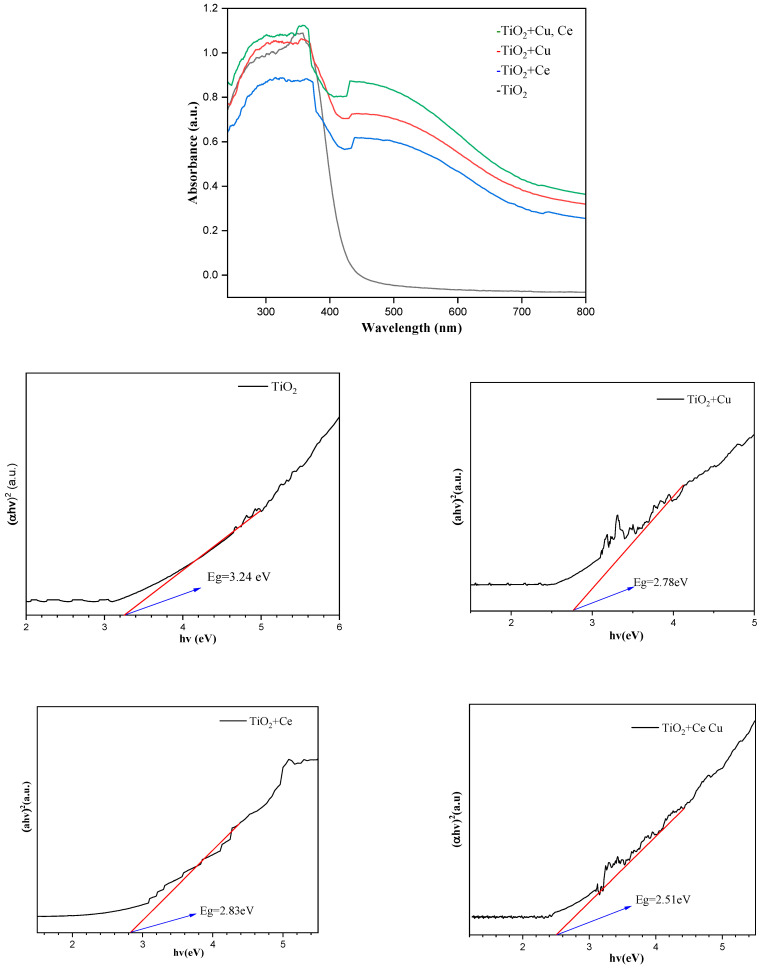
UV-Vis DRS of different metals mixed with TiO_2_ (TiO_2_ is undoped, TiO_2_ + Ce is doped with Ce, TiO_2_ + Cu is doped with Cu, TiO_2_ + Ce Cu is doped with Ce and Cu) and their Tauc plot.

**Figure 14 ijms-25-10253-f014:**
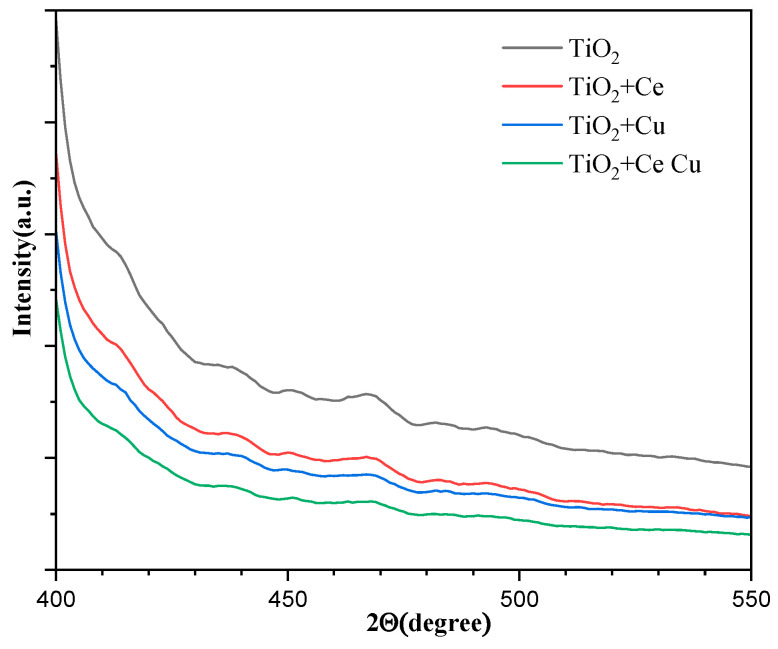
Photoluminescence spectra of TiO_2_-supported metals Cu and Ce (TiO_2_ is undoped, TiO_2_ + Ce is doped with Ce, TiO_2_ + Cu is doped with Cu, TiO_2_ + Ce Cu is doped with Ce and Cu).

**Figure 15 ijms-25-10253-f015:**
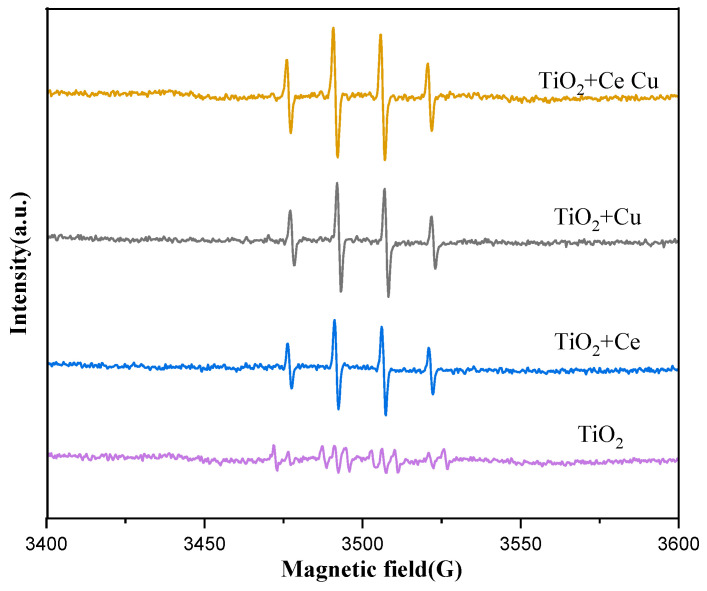
EPR spectra of TiO_2_-supported metals Cu and Ce (TiO_2_ is undoped, TiO_2_ + Ce is doped with Ce, TiO_2_ + Cu is doped with Cu, TiO_2_ + Ce Cu is doped with Ce and Cu).

**Figure 16 ijms-25-10253-f016:**
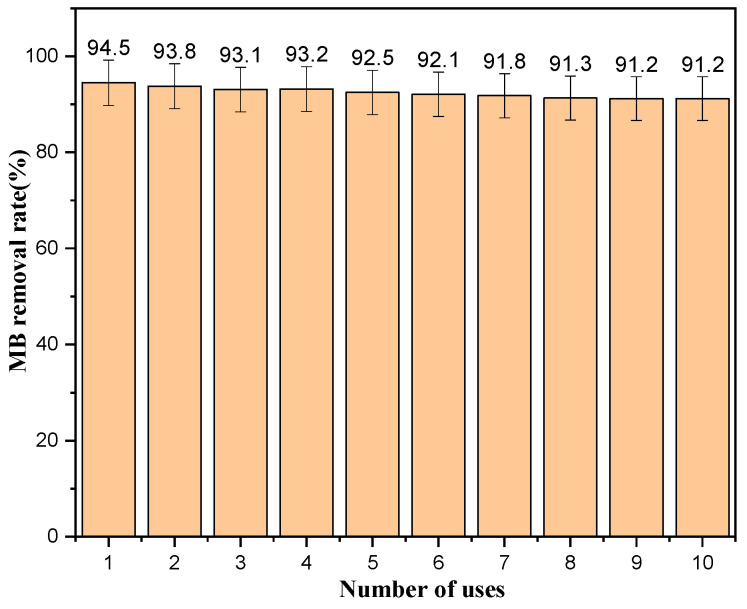
Degradation rate of catalyst after repeated use.

**Figure 17 ijms-25-10253-f017:**
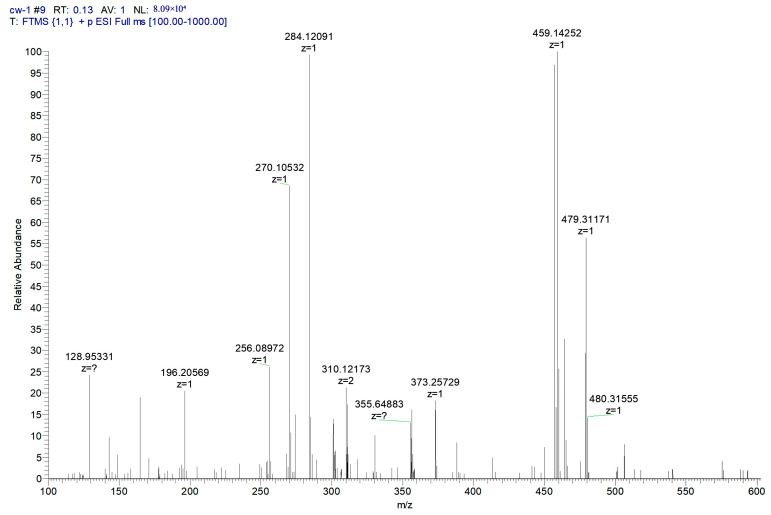
LC-MS spectrum showing compounds identified in the MB solution after photocatalytic oxidation. ? indicates uncertainty about the valence of the mass to charge ratio.

**Table 1 ijms-25-10253-t001:** Specific surface area of catalyst with different molecular weights.

Molecular Weight of PEG	Specific Surface Area (m^2^·g^−1^)
400	261.5
800	297.5
1000	329.7
1500	279.6
2000	272.6

**Table 2 ijms-25-10253-t002:** Specific surface area of catalyst with different doped metals.

Doped Metal	Specific Surface Area (m^2^·g^−1^)
Undoped	329.7
Cu	311.3
Ce	430.2
Cu and Ce	337.2

**Table 3 ijms-25-10253-t003:** Mass spectrometry analysis of MB after photodegradation.

Structural Formula	Molecular Formula	Molecular Weight
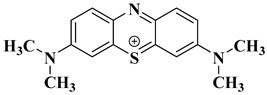	C_16_H_18_N_3_S^+^	284.1209
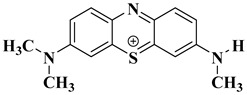	C_15_H_16_N_3_S^+^	270.1054
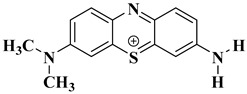	C_14_H_14_N_3_S^+^	256.0898

**Table 4 ijms-25-10253-t004:** Main experimental reagents.

Compound Name	Chemical Formula	Manufacturer
Tetrabutyl titanate	Ti(OC_4_H_9_)_4_	Tianjin Damao Chemical Reagent Factory (Tianjin, China)
Ethyl orthosilicate	(C_2_H_5_O)_4_Si	Tianjin Damao Chemical Reagent Factory
Cerium nitrates hexahydrate	Ce(NO_3_)_3_·6H_2_O	Shanghai Maclean’s Biochemical Technology Co., Ltd. (Shanghai, China)
Copper nitrate trihydrate	Cu(NO_3_)_2_·3H_2_O	Shanghai Maclean’s Biochemical Technology Co., Ltd.
Methylene blue	C1_6_H_18_ClN_3_S·3H_2_O	Fuchen (Tianjin) Chemical Reagent Co., Ltd. (Tianjin, China)
N-butanol	C_4_H_10_O	Tianjin Damao Chemical Reagent Factory
Hexadecyltrimethylammonium bromide	C_19_H_42_BrN	Shanghai Aladdin Biochemical Technology Co., Ltd. (Shanghai, China)
Polyethylene glycol 400	HO(CH_2_CH_2_O)_n_H	Tianjin Bailun Biotechnology Co., Ltd. (Shanghai, China)
Polyethylene glycol 800	HO(CH_2_CH_2_O)_n_H	Shanghai Aladdin Biochemical Technology Co., Ltd.
Polyethylene glycol 1000	HO(CH_2_CH_2_O)_n_H	Shanghai Aladdin Biochemical Technology Co., Ltd.
Polyethylene glycol 1500	HO(CH_2_CH_2_O)_n_H	Shanghai Aladdin Biochemical Technology Co., Ltd.
Polyethylene glycol 2000	HO(CH_2_CH_2_O)_n_H	Shanghai Aladdin Biochemical Technology Co., Ltd.
Sodium hydroxide	NaOH	Beijing Chemical Factory Co., Ltd. (Beijing, China)
Anhydrous ethanol	C_2_H_6_O	Beijing Chemical Factory Co., Ltd.

**Table 5 ijms-25-10253-t005:** Main experimental instruments and equipment.

Experimental Instrument Name	Instrument Model	Producer
Electronic balance	AL204	Mettler-Toledo International Trading (Shanghai) Co., Ltd. China (Shanghai, China)
CNC ultrasonic cleaner	KQ-400DB	KunShan Ultrasonic Instruments Co., Ltd., China (Kunshan, China)
1650 °C atmosphere tube furnace	MR13	Beijing Huace Testing Instrument Co., Ltd., China (Beijing, China)
Electric blast constant-temperature drying oven	101-0B	Shaoxing Yuanmore Machine and Electrical Equipment Co., Ltd., China (Shaoxing, China)
Field emission scanning electron microscope	S-4700	Hitachi, Tokyo, Japan
Specific surface area analyzer	ASAP 2460 2.02	Micromeritics, Norcross, GA, USA
X-ray photoelectron spectrometer	ESCALAB 250	ThermoFisher Scientific, Waltham, MA, USA
UV spectrophotometer	L6S	Shanghai Yidian Scientific Instrument Co., Ltd., China (Shanghai, China)
Fluorescence spectrometer	F-7000	Hitachi, Tokyo, Japan
X-ray diffractometer	XRD-6000	Shimadzu, Kyoto, Japan
Simultaneous thermal analyzer	TGA/DSC 3+	Mettler-Toledo, Zurich, Switzerland

## Data Availability

The data presented in this study are available upon request from the corresponding author.

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
