# Peer review of "Preparation and Photodegradation of TiO2 Thin Films on the Inner Wall of Quartz Tubes"

_ijms, 2024, doi:10.3390/ijms251910253_

Round 1

Reviewer 1 Report

Comments and Suggestions for Authors

This study examined tetrabutyl titanate was used as the starting material, n-butanol as the solvent, polyethylene glycol as the pore-forming agent, tetraethyl orthosilicate as the cross-linking agent, and hexadecyl ammonium bromide as the surfactant. The effects of thickness, specific surface area, calcination temperature, and loading metal concentrations are studied. Overall, the manuscript is still premature and weak in innovation, so I do not think that this manuscript is qualified for publication in International Journal of Molecular Sciences. Below is a summary of these issues:

1. Abstract is a short summary of the research paper, the author spent too much texts introducing the research background and did not highlight the innovative points.

2. The title of this article emphasizes that the TiO2 film grows inside the quartz tube, but this part is not mentioned in the introduction.

3. The figure of the photocatalytic degradation mechanism has no figure caption.

4. what is the bulge in the figure 2-1a?

5. Line 92 “However, when the TBT concentration reached 10%, the growth rate of the TiO2 films decreased significantly”

The growth rate has not been discussed before. How to judge that the growth rate has decreased?

6. Figure 2-2, the author must mark the value of film thickness in the images.

7. What is the reason for the increased TiO2 thickness to improve catalytic efficiency?

8. There are no characterization or references to support the theory in 2.1.2 section.

9. In 2.1.4, the authors should give the specific surface area of the undoped film.

10. Each peak in XRD patterns must be labeled in figure 2-6a.

11. The authors need to perform TEM characterization to delve into the pattern and distribution of metal doping.

12. If the doping level is less than 5%, EDS cannot be used for quantitative analysis.

Comments on the Quality of English Language

1. Grammatical and logical errors. Line 105 “however, the photodegradation efficiency did not increase with the thickness of the TiO2 film, resulting in an increase in the degradation efficiency.”

2. The language of the paper is too colloquial and needs to be improved. Line 102 “We can see that*******”,

Reviewer 2 Report

Comments and Suggestions for Authors

This manuscript describes the synthesis of titanium dioxide film on the inner wall of quartz tubes using in situ sol–gel method. The preparation conditions and their effect on the photodegradation of MB were optimized. The manuscript is well written. Albeit the topic is of great interest, a few concerns should be considered before consideration for publication:

1.       The abstract has to be rewritten and improved, focusing on the research question and the gap it fills, methods and main findings. No need for naming the chemicals in the abstract.

2.       Authors should highlight in the end of the introduction what research gap does this research address and its novelty.

3.       The figures should be renumbered as 1, 2, 3, etc

4.       The pore size can be obtained from the BET analysis and would support the argument of effect of PEG on porosity.

5.       SEM of the doped TiO2 should be included to show its surface modification and metal distribution on TiO2.

6.       The introduction should be strengthened by considering recent works on photocatalytic nanomaterials for photodegradation e.g. 10.1016/j.surfin.2024.104566, 10.1016/j.carbon.2023.118579 and photoelectrochemical processes of TiO2 e.g. Int. J. Electrochem. Sci 7 (2012) 3610-3626, J Nanobiotechnol 19, 340 (2021)

7.       In the manuscript it stated “co-doping with Cu and Ce of TiO2 reduces the bandgap width, expands the absorption range of visible light”. A reference for this should be added.

8.       The control of TiO2 thin film properties via varying the synthesis and calcination conditions should be referenced by this work (Nanotube Arrays as Photoanodes for Dye Sensitized Solar Cells Using Metal Phthalocyanine Dye)

9.       The rate constant of MB degradation can be calculated based on the degradation efficiency data by referring to other works as (ZnO-Bi2O3 Heterostructured Composite for the Photocatalytic Degradation of Orange 16 Reactive Dye….and Fagonia arabica extract-stabilized gold nanoparticles as a highly selective colorimetric nanoprobe for Cd2+ detection and as a potential photocatalytic…)

10.   It seems from Fig. 9 that doping with Cu or Ce didn’t increase the degradation efficiency compared to undoped TiO2. Any explanation? And what is the advantage then of the doping?

11.   In XPS, the fitting doesn’t match well the experimental data especially for Ce spectra and the baseline of Cu spectra should be corrected.

12.   It is advised to included error bars for the results in figure 9, 14 to represent reproducibility.

13.   A comparison of the degradation efficiency and time with other relevant materials should be included.

Comments on the Quality of English Language

minor

Round 2

Reviewer 1 Report

Comments and Suggestions for Authors

The author has carefully revised the paper, which is now more rigorous and ready for publication.

Author Response

Please find attached the reviewer's comments.

Reviewer 2 Report

Comments and Suggestions for Authors

Authors have addressed most of the comments, however a few comments from the previous report weren't considered and should be considered in this work before acceptance:

1. The rate constant of MB degradation can be calculated based on the degradation efficiency data by referring to other works as (ZnO-Bi2O3 Heterostructured Composite for the Photocatalytic Degradation of Orange 16 Reactive Dye….and Fagonia arabica extract-stabilized gold nanoparticles as a highly selective colorimetric nanoprobe for Cd2+ detection and as a potential photocatalytic…)

2. The literature on photocatalytic nanomaterials for photodegradation should be updated by adding this recent work 10.1016/j.surfin.2024.104566 instead of this 10.1016/j.carbon.2023.118579

3. The figures should be renumbered as Figure 1, 2 and not Figure 2.1, Figure 2.2. Authors should comply with the journal's format. Look at recent papaers published in this journal for guide.

Author Response

Please find attached the reviewer's response.

Round 3

Reviewer 2 Report

Comments and Suggestions for Authors

Authors have addressed some of the concerns. However, after evaluating the revised submission, changes made were not sufficient to meet the publication standards and several important issues that have been raised in the previous two reports remained unresolved, particularly regarding updating the literature survey and highlighting the general prospect of the work. Authors have been given the opportunity twice to revise their work, however given their reply and reputal and the overall assessment, the manuscript needs to be carefully revised and resubmitted.

1. The rate constant k of MB degradation should be calculated based on the degradation efficiency data, at least for different catalysts in Fig. 10. Authors should take guide and refer to the suggested references in the previous reports. This is a very simple graph between ln(C/Co) versus time and the slope gives the k value. There is no page limit for this journal.

2. The included UV-Vis data in Figure 12 looks unusual where the curve has a kink at about 450 nm. What is the reason? and two peaks appeard (one broad at 500 nm and another at 350 nm) that weren't discussed. What do these two peaks represent? and which of them is used for the Eg calculations.

3. The Photoluminescence spectra in Figure 13 don't show the expected peak but just a decaying line. Photoluminescence of TiO2 materials normally shows a peak aroung 450-600 nm, which is absent here. Any explanation.

4. The literature on photocatalytic nanomaterials for photodegradation should be updated and strengthened by considering recent works e.g. https://doi.org/10.1016/j.surfin.2024.104556

5. Tables should be renumbered as Table 1, 2, etc.

6. The XPS fitting lines still don't match the measured data. One can clearly see the baseline of Cu 2p spectrum isn't mathing the experimental data.

My above comments aren't meant to postopone the decision, but rather to improve the quality of the work and authors should have at least properly considered the suggestions that are easily doable, instead of excusing with page limit or not relevant references.

Author Response

The response to Reviewer 2's third review suggestion is attached.

Round 4

Reviewer 2 Report

Comments and Suggestions for Authors

Authors have been given the opportunity thrice to carefully revise their work, however given their reply and reputal and the overall assessment, changes made were not sufficient to meet the publication standards and several important issues that have been raised in the previous three review reports were ignored.

Here are some of my comments:

1. The rate constant k is know as an important paramter to evaluate the degradation of dyes and performance of photocatalysts. This is a common standard in the field and authors igonored to solve it although they already have the data. There is no page limit for this journal as the authors claim in their previous reply.

2.  Two peaks appeard in UV-Vis spectra (one broad at 500 nm and another at 350 nm) that weren't discussed. What do these two peaks represent? and which of them is used for the Eg calculations. After a simple search, one can find that such kind of materials always show a bell-like form. How do authors then exclude that the observed change is just a baseline alteration.

3. Discussions and interpretations aren't supported by robust and reliable data. Authors have answered in their last reply repot that "Baseline fluctuations can affect data reliability". This makes the reviewer doubts the reliability and reproducibility of the data.

4. XPS data are misinterpreted and the given numbers in the manuscript aren't represented in the spectra. For example, authors stated "Cu peaks at 932.6 eV are 932.6 eV, 932.4 eV, and 933.6 eV, indicating that Cu mainly exists in the forms of Cu0, Cu+, and Cu2+" There are no sub-peaks shown in the spectra of Cu 2p 3/2. The same applies for Ce spectrum. In addition, the fitting of Cu 2p spectrum doesn't match the discussion and claimed observations.

5. references weren't updated.

Author Response

Please refer to the attachment for the response to the review comments.
